# Optimal inventorying and monitoring of taxonomic, phylogenetic and functional diversity

**Pedro Cardoso**[1,2]*, **Miquel A. Arnedo**[3], **Nuria Macías-Hernández**[2,4], **William D. Carvalho**[5,6,7,8], **José C. Carvalho**[1], **Renato Hilário**[8,9]

1 Centre for Ecology, Evolution and Environmental Changes (cE3c) & CHANGE—Global Change and Sustainability Institute, University of Lisbon, Lisboa, Portugal, 2 Laboratory for Integrative Biodiversity Research (LIBRe), Finnish Museum of Natural History (Luomus), University of Helsinki, Helsinki, Finland, 3 Department of Evolutionary Biology, Ecology & Environmental Sciences, and Biodiversity Research Institute (IRBio), Universitat de Barcelona, Barcelona, Spain, 4 Department of Animal Biology, Edaphology and Geology, University of Laguna, La Laguna, Canary Islands, Spain, 5 Facultad de Ciencias, Departamento de Ecología, Terrestrial Ecology Group (TEG-UAM), Universidad Autónoma de Madrid, Madrid, Spain, 6 Centro de Investigación en Biodiversidad y Cambio Global (CIBC-UAM), Universidad Autónoma de Madrid, Madrid, Spain, 7 Associação Mata Ciliar, Jundiaí, Brazil, 8 Programa de Pós-Graduação em Biodiversidade Tropical, Universidade Federal do Amapá (UNIFAP), Macapá, Brazil, 9 Department of Environment and Development, Laboratory of Ecology, Federal University of Macapá, Macapá, Brazil

* pedro.cardoso@helsinki.fi

**Data Availability Statement:** All code to replicate the analyses is deposited at https://github.com/cardosopmb/optimSampling Spider distribution data are fully available in https://doi.org/10.3897/BDJ.6.e29443 Genetic data are fully available in

## Abstract

Comparable data is essential to understand biodiversity patterns. While assemblage or community inventorying requires comprehensive sampling, monitoring focuses on as few components as possible to detect changes. Quantifying species, their evolutionary history, and the way they interact requires studying changes in taxonomic (TD), phylogenetic (PD) and functional diversity (FD). Here we propose a method for the optimization of sampling protocols for inventorying and monitoring assemblages or communities across these three diversity dimensions taking sampling costs into account. We used Iberian spiders and Amazonian bats as two case-studies. The optimal combination of methods for inventorying and monitoring required optimizing the accumulation curve of α-diversity and minimizing the difference between sampled and estimated β-diversity (bias), respectively. For Iberian spiders, the optimal combination for TD, PD and FD allowed sampling at least 50% of estimated diversity with 24 person-hours of fieldwork. The optimal combination of six person-hours allowed reaching a bias below 8% for all dimensions. For Amazonian bats, surveying all the 12 sites with mist-nets and 0 or 1 acoustic recorders was the optimal combination for almost all diversity types, resulting in >89% of the diversity and <10% bias with roughly a third of the cost. Only for phylogenetic α-diversity, the best solution was less clear and involved surveying both with mist nets and acoustic recorders. The widespread use of optimized and standardized sampling protocols and regular repetition in time will radically improve global inventory and monitoring of biodiversity. We strongly advocate for the global adoption of sampling protocols for both inventory and monitoring of taxonomic, phylogenetic and functional diversity.

https://doi.org/10.6084/m9.figshare.26236652.v2
Functional data are fully available in https://doi.org/
10.6084/m9.figshare.8320004.v3 Bat diversity data
are fully available in https://doi.org/10.1007/
s42974-022-00131-5.

**Funding:** PC has received funding from the
European Union's Horizon 2020 and Horizon
Europe research and innovation programmes
under grant agreements No. 861924 (SustInAfrica)
and 101081964 (BioMonitor4CAP) respectively. PC
was also supported by the Centre for Ecology,
Evolution and Environmental Changes (cE3c,
https://doi.org/10.54499/UIDB/00329/2020) and
the Global Change and Sustainability Institute
(CHANGE, https://doi.org/10.54499/la/p/0121/
2020). MA was supported by grant 485/2012 of
the Organismo Autónomo de Parques Nacionales
(Ministerio de Agricultura, Alimentación y Medio
Ambiente). NMH was supported by the H2020-
MSCA-IF-2015 Programme (BIODIV ISLAND-
CONT project, Grant no. 706482). WDC is
supported by 'Ayudas Maria Zambrano' (CA3/
RSUE/2021-00197), funded by the Spanish
Ministry of Universities. Open access funded by
Helsinki University Library.

**Competing interests:** The authors have declared
that no competing interests exist.

## Introduction

The current level of biodiversity loss is above anything experienced in the near past, with our era being recognized as the grand stage to the sixth extinction era [1,2]. Yet, useful knowledge for conserving most species is extremely scarce and the current population trends for most taxa are unknown [3,4]. Currently, information about biodiversity is highly biased both geographically and taxonomically [3,5,6]. Improving our knowledge of global biodiversity and assessing its changes is a huge challenge. Therefore, there is a need to optimize sampling protocols to better allocate sampling effort and resources [7,8].

### Inventorying or monitoring?

The answer to this question is most of the time obvious. Although many researchers and practitioners tend to see these tasks as equivalent or interchangeable, the processes and goals are essentially different. An inventory is required if the goal is to know which species, clades or traits occur where, to study community assemblage patterns and its drivers or to prioritize areas for conservation, among many other uses that require comprehensive knowledge of community composition. As the aim is to maximize sampled α-diversity, all taxa of interest should be targeted, a high level of sampling completeness is needed, and sampling should cover as much as possible the region and seasons of interest (Table 1). On the other hand, this is either a one-time or a rare event that given the effort involved is almost invariably impossible to replicate in the short or even medium-term for all but species-poor taxa.

In contrast, monitoring is required if the goal is to detect, and if possible quantify, changes in community composition (or of a single species of particular interest) in space or, more often, over time [9–15]. It should be noted that monitoring might involve other parameters, including abiotic variables and species networks, which are outside the scope of this work. To successfully detect changes in community composition, i.e. β-diversity, one should, at any sampling occasion, minimize differences between sampled and true (or estimated) β-diversity. To achieve this, one may focus on selected taxa, which means that completeness is not required, and sampling can cover only a subset of sites sampled in the height of seasonal abundance for the selected taxa (Table 1). This minimizes the required time and resources and allows frequent replication. In other words, if inventorying often requires comprehensive, time-consuming methods, monitoring should build on it and focus on as few components as possible that still allow the correct perception of changes on predetermined indicators in time, or often, space (Table 1) [10,11]. To sample or acquire useful information for both purposes can therefore be viewed as fundamentally different but complementary processes, hereafter called α and β-sampling (Table 1).

**Table 1. Characteristics of inventorying and monitoring of biodiversity.**

|  | Inventorying | Monitoring |
|---|---|---|
| **Target** | Maximize sampled α-diversity | Minimize difference between sampled and true β-diversity |
| **Sampling effort** | Major | Minor |
| **Geographic scope** | Global | Often of selected sites due to limited resource availability |
| **Seasonality** | Full year | Selected season in the case of variation (often of maximum diversity) |
| **Periodicity** | Single or sporadic (> 10 years) | Regular (1 to 10 years) |

Both inventorying and monitoring of biodiversity require defining what biodiversity is in the first place. Its usual meaning covers the variety of species, their evolutionary history and how they interact to constitute complex assemblages of communities and ecosystems. Quantifying all these aspects leads to the need to study three different facets of biodiversity: taxonomic (TD), phylogenetic (PD), and functional (FD) diversity. If TD covers species composition and their abundances, PD also quantifies the evolutionary history behind a given community and FD quantifies the number and variety of traits and functions the different species perform in their ecosystems [16]. The existence of optimized protocols to inventory and monitor TD, PD and FD of a much wider range of taxa is therefore essential if we really want to follow the current trends in biodiversity as a whole, from local to global scales [8]. Several global initiatives are currently underway for the inventory and monitoring of biodiversity [17]. Monitoring has however been the main concern, given the current rates of biodiversity loss. Some of the main users of monitoring data are the GEO BON—Group of Earth Observation Biodiversity Observation Network, within which Pereira et al. [18] proposed a global terrestrial species monitoring program, advocating that vascular plants and birds should be the candidate taxa for that purpose. Pereira et al. [18] noted that Essential Biodiversity Variables relating to species, genes or traits require representative sampling across taxonomic groups, recognizing the difficulty in obtaining such data. The GEO BON set of variables to follow is now expanding to include other taxa such as butterflies [19]. Furthermore, the IUCN (International Union for the Conservation of Nature) proposed the Red List Index (RLI), which applies to any organism group, yet only calculated for a handful of taxa [20,21]. Finally, the Living Planet Index (LPI) is based only on vertebrate population trends, limiting its application and interpretation to most of the global biodiversity, i.e., invertebrates and plants [22]. All these global initiatives have several points in common. They do not currently consider most taxa, and there is no evidence that the ones covered can serve as good surrogates for detecting trends in the remaining biodiversity [3].

Cardoso and Leather [8] proposed a general framework to eliminate biases and optimize biodiversity inventorying and monitoring. Also, Cardoso [23] proposed a method to optimize the inventory of species (TD) (COBRA protocol: Conservation Oriented Biodiversity Rapid Assessment), particularly focusing on mega-diverse groups and using spiders as a case study. This former study demonstrated that it is possible to sample in a standardized, yet optimized way, and the protocol proposed just a decade ago is now being used worldwide [24–27]. Here we extend the concept from inventorying to monitoring and its application from TD to PD and FD. We also propose how to take into account different costs, both fixed and variable, incurred by different sampling methods. We demonstrate how to optimize sampling protocols for both invertebrates (Iberian spiders) and vertebrates (Amazonian bats).

## Materials and methods

### Basic algorithm for inventorying (α-sampling)

Any sampling event has two kinds of costs. Fixed costs imply some kind of initial investment, such as buying traps or paying trips. Variable costs add up with the sampling effort, for example paying person-hours of field or lab work or buying extra tubes and reagents to store and process samples. All these should be taken into account for any cost-benefit analysis. In addition, we can build on a predetermined set of samples to improve their efficiency with complementary methods. This constrained algorithm may be useful when we want to improve on an existing sampling protocol, guaranteeing a minimum common denominator to allow comparisons between old and new protocols. Another case is when the habitat complexity changes considerably in time and the old methods are no longer optimal. One might optimize sampling

effort for the most complex situation, but simplify the protocol when needed, guaranteeing future comparability. Improving a protocol based on preexisting samples implies forcing the optimization exercise to start with such constraints and searching for the methods that better complement it when adding further samples (see for example [25]).

To reach the optimal combination of methods for any sampling effort we propose to minimize the bias of sampled diversity (TD, PD or FD) for any undersampling level compared with true (or often estimated) diversity as revealed by intensive sampling, for both inventorying and monitoring. Here we define bias as the difference between observed and true diversity (for inventorying, diversity is species richness, for monitoring it is beta diversity). This requires information on (near-)complete inventories for multiple sites using multiple methods. Once these data are available for any taxon, habitat or region, the following steps are:

1. Calculate fixed and variable costs per method and sample;

2. Start with a pool of samples per method. This is often 0 for all samples (an empty pool), but one might want to build on previous sampling protocols;

3. Create all possible combinations of samples per method that include the pool;

4. Calculate the cost of each combination;

5. Per combination, repeat multiple times (we used and recommend at least 1000):

   a. Without replacement, randomly simulate a sampling event with the defined number of samples per method.

   b. Calculate the absolute difference between sampled and true (or estimated) diversity (i.e. bias);

6. Per cost value choose the combination with the least bias.

One might also want to create a sequence of nested combinations, this way allowing for flexibility depending on the level of completeness required and resources available. In this case, in step 3 only combinations that add one sample per method are chosen and the new pool will be made up by the samples already chosen plus one sample of the method that provides the steepest slope to the accumulation curve (i.e. highest gain per cost unit).

## Extending to multiple dimensions and sites

We propose to optimize the sampling of PD and FD in addition to TD. Many ways exist to quantify PD [28] and FD [29], each with advantages and disadvantages, beyond the scope of this work. The principle is the same and intended to minimize the differences between sampled and true (or estimated) diversity, almost invariably taxonomic, phylogenetic or functional richness.

In addition, one should not base any sampling protocol on the experience of a single site or season. To make results comparable between these, we propose to standardize richness values between 0 and the true (or estimated) richness of each event, this way all values across sites or seasons vary between 0 and 1. This allows using multiple sampling events in a single analysis where all events have equal weight for the optimal protocol.

## Optimization of monitoring protocols (β-sampling)

While the basic algorithm was described for inventorying, hence comparisons were made for richness, i.e., α-diversity, the same principle can be applied for monitoring, i.e., detecting changes in community composition in space and time using a β-diversity framework [30].

Optimal monitoring requires minimizing the difference between sampled and true (or estimated) β-diversity. To guarantee such a goal, one can use any β-diversity measure of interest. We propose to use a framework that disentangles the two antithetic processes that drive overall β-diversity: species replacement and species richness differences [31]. This framework was further expanded to PD and FD [32] and can be represented as:

$$\beta_{total} = \beta_{repl} + \beta_{rich}$$

where $\beta_{total}$ represents the total community variation, $\beta_{repl}$ gives its fraction resulting from species replacement, i.e., the substitution of species across sites or over time, and $\beta_{rich}$ accounts for the variation as a result of species richness differences, determined by the net loss / gain of species.

These measures of beta diversity are particularly robust to biases due to undersampling [33] and therefore the sampling effort necessary to attain low bias is small compared with α-sampling. It is also a necessary property of any monitoring protocol to be easily replicated in time, so the overall number of samples could never be as high as for inventory.

The target for the algorithm is to minimize the summed absolute difference between sampled and true (or estimated) β-diversity values using all three measures for TD, PD and FD. To make results fully comparable with alpha diversity, we propose to measure the efficiency of a combination of samples as 1—bias, where bias is the mentioned summed difference.

It should be noted that monitoring as defined here refers to quantifying changes of community composition in space or time. Many monitoring programs target single species abundances, for example of threatened or indicator species; or other variables such as pollution or other abiotic factors. The reader should take our goal into account for the interpretation of the methods and final sampling protocols.

## Case-studies

In the present work, we use Iberian spiders and Amazonian bats as case-studies. The ecological differences among these groups evidence that the same optimization procedure could be applied to any taxon that requires a combination of sampling methods to be fully represented, from plants and fungi to mammals. For Iberian spiders, which are a highly speciose group, we considered that an inventory should reach at least 50% of the estimated diversity. For Amazonian bats, we considered that a successful inventory should reach at least 90% of the projected diversity. For monitoring, we considered that bias should be lower than 10% in all cases. These values can be adapted to any case and are only for demonstration purposes. It should be noted that monitoring is usually done using data from several years. In this case we used a space-for-time approach for both case-studies, simulating a situation where habitats are converted from one habitat to another in time using spatial surrogates.

## Iberian spiders

Besides high global diversity with over 52000 known spider species [34], local diversity can also be extremely high, with several hundreds of species co-existing in areas as small as 1 ha, especially in high-diversity biomes such as tropical forests [35].

Our team sampled Iberian spiders in 18 sites using a common protocol since 2004 (S1 Table). In two forest sites, Arrábida and Gerês, sampled in 2004 and 2005 respectively, we collected a much larger number of samples (320) than in all others (24) and these were used previously to define the current inventory protocol for TD (COBRA) [23,36,37]. The remaining 16 sites were sampled between 2013 and 2014 mostly using 24 samples only, intended to capture 50% of the estimated species richness [26]. Numerous habitat types were sampled using a

combination of methods: aerial searching, beating trees and branches, ground searching, sweeping and pitfall trapping (for details of the sampling methods see [36,37]). All samples (except pitfall traps) comprised one-hour of active sampling, measured with a stopwatch and were performed both during days and nights. Diurnal and nocturnal samples of each method were considered different methods as assemblages assessed are considerably different. Pitfall traps (48) were pooled in groups of four as we estimated that four traps take an average of one hour to set up, so that they were comparable with other methods in terms of sampling effort. In this case, because fixed costs are very low and variable costs per method similar, we considered that there were no fixed costs, and variable costs were one per sample (hour) for every method. We also did a nested optimization, where a single sample was added to the pool at each step, according to the method that was found to be most efficient.

### Amazonian bats

Mist nets and acoustic recorders are the most used and efficient methods for sampling bats [38]. However, each of these methods is biased toward different families within Chiroptera; Phyllostomidae bats being more captured by mist nets and aerial insectivorous bats (e.g., Vespertilionidae and Molossidae) being more recorded using acoustic recorders [38,39]. Therefore, to inventory or monitor the order Chiroptera, these two methods are considered complementary [39,40] and we used both in a landscape composed of savannahs and forest patches in Northeastern Amazon. Each of the 12 sampled sites were surveyed for 2 non-consecutive nights during the rainy season. We waited at least 30 days before resampling the same site, thus avoiding net shyness [41], although researchers can also move the nets some meters away to allow surveying on consecutive nights. In each survey night, we installed 18 mist-nets which remained open for six hours from 15 minutes before sunset, being nine mist-nets in the savannas and nine in the forest patches, making two 110 m transects. In each of these transects, we installed three Audimoth acoustic recorders, which were set to record continuously for six hours from 15 minutes before sunset, with a sampling rate of 384 kHz and medium gain (see [39] for details).

In this case study, we considered the costs of implementing these methods. The 18 mist-nets used in the study cost $360 ($20 each), and the 20 aluminum poles used to install the mist-nets cost $160 ($8 each). Therefore, mist-net surveys have a fixed cost of $520, i.e. the cost of applying this method independently of the number of survey nights. However, mist-netting also has a variable cost, related to the number of nights in which the method is implemented. For each mist-net survey night we needed four researchers in the field, which represented a cost of $192 per night (i.e. $48 per researcher). Surveying a single site costs $904 ($520 + 2 nights x $192) and each additional site included in the sampling costs another $384.

For surveying with acoustic recorders, we have a fixed cost of $883.20, given that we used six Audiomoth acoustic recorders ($60, each), six 32 Gb storage cards ($67.20, each), and one 4 Tb hard disk to store the recordings ($120). The variable cost for surveying each site with acoustic recordings is $1062: $6 for plastic bags to protect the recorders, $192 for per-diem of two researchers to install and retrieve the recorders, and $864 of salary of one researcher to analyze and identify the acoustic records ($48 of per-diem for an expert in identifying bat vocalizations and 3 days to analyze the records of each of the 6 recorders deployed per site, totalling 18 days of work per site).

To evaluate the optimal protocol for α-sampling, we considered the number of sites to survey with each method. Because β-sampling involves the comparison of a number of sites, to evaluate the optimal protocol for β-sampling, we considered whether we should use mist-nets or acoustic recorders in all sites altogether.

## Phylogenetic and functional diversity

PD and FD were measured as the sum of the length of branches on a phylogenetic or functional tree, respectively [32,42–44]. Since TD can also be represented by a tree with each taxon linked directly to the root by a branch of unit length (star tree), tree diagrams provide a common basis for sampling optimization of TD, PD and FD [32,44,45].

For Iberian spiders, a phylogenetic tree was built using the procedure detailed in Macías-Hernández et al. [46]. In short, a concatenated matrix of the mitochondrial cytochrome c oxidase subunit I (COI) and the nuclear 28S rRNA (28S) genes sequenced for the 372 species collected in the 16 sites was constructed (see further details on laboratory protocols and molecular data in [46]). Approximately 1000 additional species representing most spider families, sampled for the same two gene regions, together with four additional genes (12S rRNA,16S rRNA, histone H3 and 18S rRNA), available from a previous study [47], were added to the COI and 28S concatenated matrix. Then, a Maximum Likelihood (ML) phylogenetic tree was inferred using the program RAxML-HPC v. 8.2.12 [48], remotely run on XSEDE at the CIPRES Science Gateway [49]. The phylogenetic analysis was performed with a topological constraint provided by a phylotranscriptomic study on spiders [50], which also provided the calibration points to infer time-stamped phylogeny. The resulting ML tree was made ultrametric using the *congruify.phylo* function in the R package *GEIGER* [51], and the resulting table of calibration points was used to date the target trees using the program treePL [52]. This tree was used for the optimization of monitoring PD.

In lieu of precise phylogenetic information on the member taxa of a community, PD can also be calculated from an ultrametric tree representing the Linnaean hierarchy [53]. As no genetic data were available for the two sites used for the optimization of inventory, a tree based on taxonomic hierarchy was constructed as a surrogate using the function *linnean* in the R package *BAT* [45]. This tree was used for the optimization of inventorying PD.

For Amazonian bats, we obtained an ultrametric consensus tree from the 1000 trees retrieved from Upham et al. [54] using the R package *ape* [55]. This phylogeny was based on a 31-gene supermatrix and included all the 48 species detected in the study.

For Iberian spiders, we constructed a functional tree using data from 11 functional traits, namely: mean total body size (males and females), prosoma length, prosoma width, prosoma height, tibia I length, fang length, dispersal ability (ballooning propensity: (F) frequent; (O) occasional or (R) rare), vertical stratification (from epigean to arboreal), circadian activity (nocturnal / diurnal), foraging strategy (no web / web builders: capture web, sensing web, tube web, sheet web, space web and orb web), and trophic specialization (specialist or generalist), based on bibliographic searches, personal knowledge of species or derived from our own sampling data (further details in [56,57]). To avoid correlation between prosoma width and other morphological variables, we calculated the residual values of each morphological variable against prosoma width. A functional dissimilarity matrix between species was constructed using Gower's distance and rescaling variables by selecting specific weights. Each morphological variable (but prosoma width) and the binary variables related to types of webs were weighted as 1/n types. The functional tree was built as a neighbor-joining tree using the function *tree.build* of the R package *BAT* [45]. Using neighbor-joining is the only representation that allows comparability with analyses of PD using phylogenetic trees, an unbiased representation of species distances, and to use the beta diversity partitioning framework for the optimization of the monitoring protocol [44].

For Amazonian bats, we constructed a functional tree based on five traits: the logarithm of body mass, forearm size, trophic level (phytophagous or animalivorous), dietary guild (frugivore, insectivore, aerial insectivore, omnivore or sanguinivore), and vertical stratification

(canopy or understory). Body mass and forearm size were obtained from individuals captured in the field, and the trophic guild was based on Giannini and Kalko [58]. Diet was based on a repository of published papers that analyzed the diet of different species of bats around the world [59]. We assessed whether the bat species prefer using the canopy or the understory through previous studies in the Amazon [60–63]. A functional tree was built the same way as for spiders.

For both taxa, we used the functions *optim.alpha* and *optim.beta* in *BAT* [45] to find the best allocation of sampling effort for inventorying and monitoring, respectively.

## Results

### Iberian spiders

A total of 272 species and 13,064 adult individuals (from two sites) were used for optimizing α-sampling, and 587 species and 18,623 adults (from 16 sites) for optimizing β-sampling, both using a nested optimization procedure. As we previously found that six hours is the maximum effort a single researcher can perform per day without fatigue influencing results [23], we looked for combinations that would add to a total number of samples multiple of six. For α-diversity, the optimal combination of 24 samples for both TD and PD was very similar (Table 2). In both cases, the proportion of diversity sampled was higher than 50%, reaching 57% for PD. The optimal protocol was close or even over the upper confidence limit for random sampling for most of the accumulation process (Fig 1). For FD, the main difference laid on the optimality of doing ground searching during the day instead of beating or sweeping. In any case, as many species are very similar in their traits, the sampled diversity with just 24 samples was very high, reaching 79% of true FD. This was confirmed by the much faster accumulation rate of FD with increasing number of samples (Fig 1). In the constrained exercise, the algorithm was able to rapidly reach method combinations and sampling levels on par with the unconstrained optimization (Table 2; Fig 1).

Optimal number of samples per method for inventorying (α-sampling) and monitoring (β-sampling) of spider taxonomic (TD), phylogenetic (PD) and functional (FD) diversity. A total of 24 and six samples were chosen respectively (see text for details). The constrained runs start with 12 or four pitfall trap samples, respectively.

For β-diversity, the optimal combination of six samples for all measures was different but with many samples in common (Table 2). In all cases, the 1—bias from true diversity was between 0.922 (for FD) and 0.935 (for TD). The optimal protocol was however not significantly different from a random one based on the inventory protocol (Fig 1). In the constrained exercise, the algorithm was able to rapidly reach method combinations and sampling levels on par with the unconstrained optimization (Table 2; Fig 1).

**Table 2. Spider sampling optimization.**

| | | Aerial | | Beating | | Ground | | Pitfall | Sweep | | Average proportion of diversity (α) or 1—bias (β) |
|---|---|---|---|---|---|---|---|---|---|---|---|
| | | Day | Night | Day | Night | Day | Night | | Day | Night | |
| **α-sampling** | **TD** | 0 | 4 | 4 | 1 | 0 | 0 | 9 | 0 | 6 | 0.507 |
| | **PD** | 0 | 3 | 3 | 2 | 0 | 1 | 9 | 0 | 6 | 0.567 |
| | **FD** | 1 | 6 | 0 | 2 | 5 | 0 | 9 | 0 | 1 | 0.786 |
| | **TD constrained** | 0 | 3 | 3 | 0 | 0 | 1 | 12 | 0 | 5 | 0.503 |
| **β-sampling** | **TD** | - | 1 | 0 | 2 | - | - | 2 | 1 | 0 | 0.935 |
| | **PD** | - | 3 | 1 | 0 | - | - | 2 | 0 | 0 | 0.924 |
| | **FD** | - | 0 | 2 | 0 | - | - | 3 | 0 | 1 | 0.922 |
| | **TD constrained** | - | 0 | 2 | 0 | - | - | 4 | 0 | 0 | 0.932 |

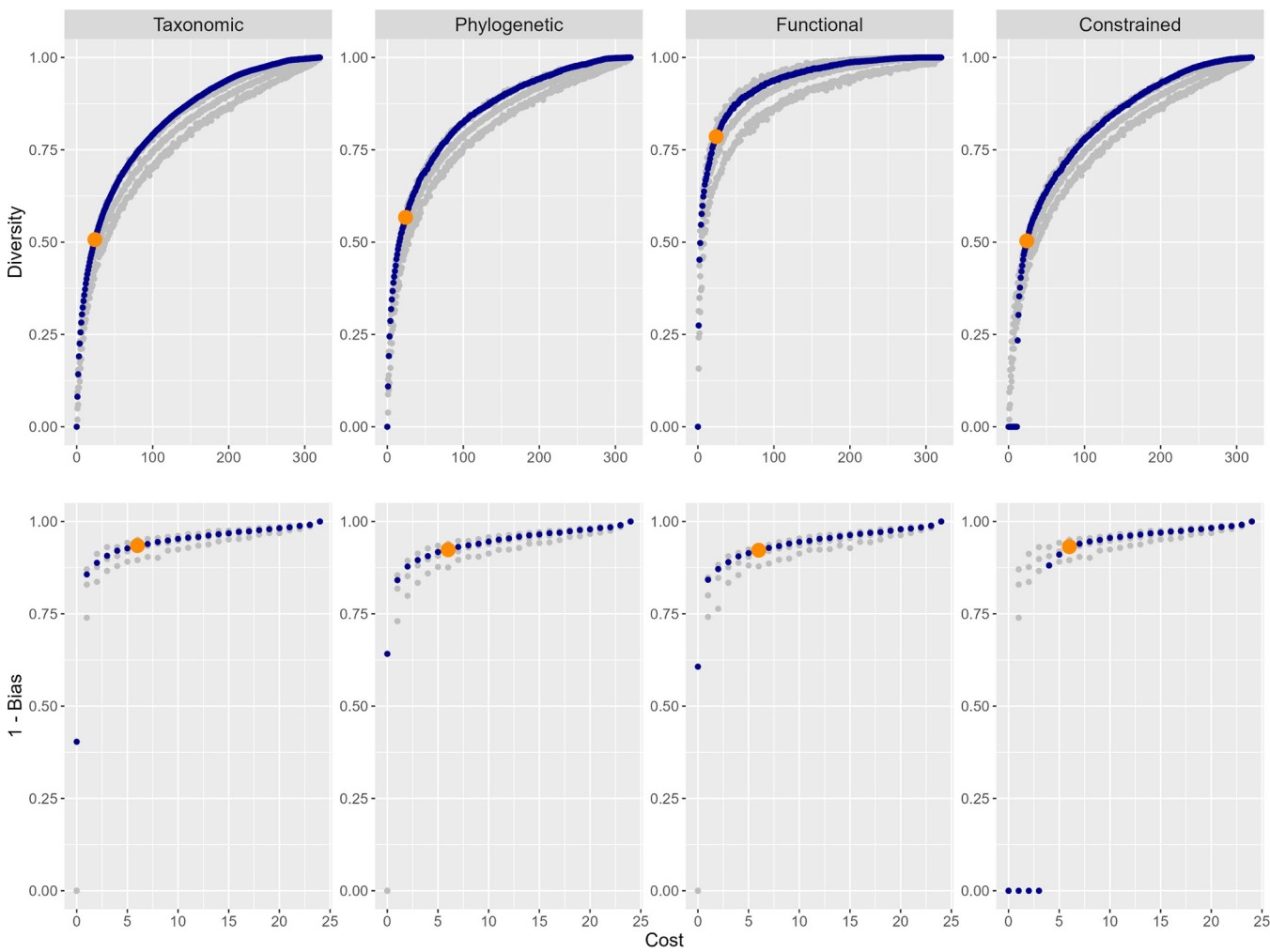

**Fig 1. Spider sampling optimization.** Accumulation curves for optimized inventorying (α-sampling; upper panels) and monitoring (β-sampling; lower panels) of taxonomic, phylogenetic, and functional diversity of spiders, plus the optimization of taxonomic diversity constrained to an initial sample of 12 and four pitfall traps, respectively, for α and β-sampling (blue dots). The results for β-sampling are presented as 1—bias, so, the greater the value, the higher the accuracy. The orange dots represent the optimal sampling protocols, i.e. with 24 and six samples, respectively (see text for details). Grey dots represent the median and 95% confidence limits of a random choice of methods along the accumulation curve.

## Amazonian bats

We captured 322 bats in mist-nets and identified 23,363 bat passes in acoustic recorders, accounting for a total of 48 bat species (38 species in mist-nets and 12 species in acoustic recorders). Despite the higher costs and lower diversity sampled by acoustic recorders, they recorded 10 unique species. Therefore, for taxonomic α-diversity, the most cost-effective combination for inventorying bats is using mist-nets in 12 sites and acoustic recorders in a single site (Fig 2). This combination returned 89.5% of the diversity with 37.7% of the cost of the full inventorying scheme. For phylogenetic α-diversity, the choice of the best solution is more subjective, given that there is a more gradual increase of the PD with the cost of implementing the methods. For example, using 5 mist-net sites and 1 acoustic recorder site results in 84.5% of the PD with 23.4% of the costs of the full scheme. Adding more mist-net sites (i.e. 10 sites) while still using a single acoustic recorder increases the recorded PD to 90.3%, but also increases the costs to 33.4% of the full scheme. With 12 mist-net sites and three acoustic

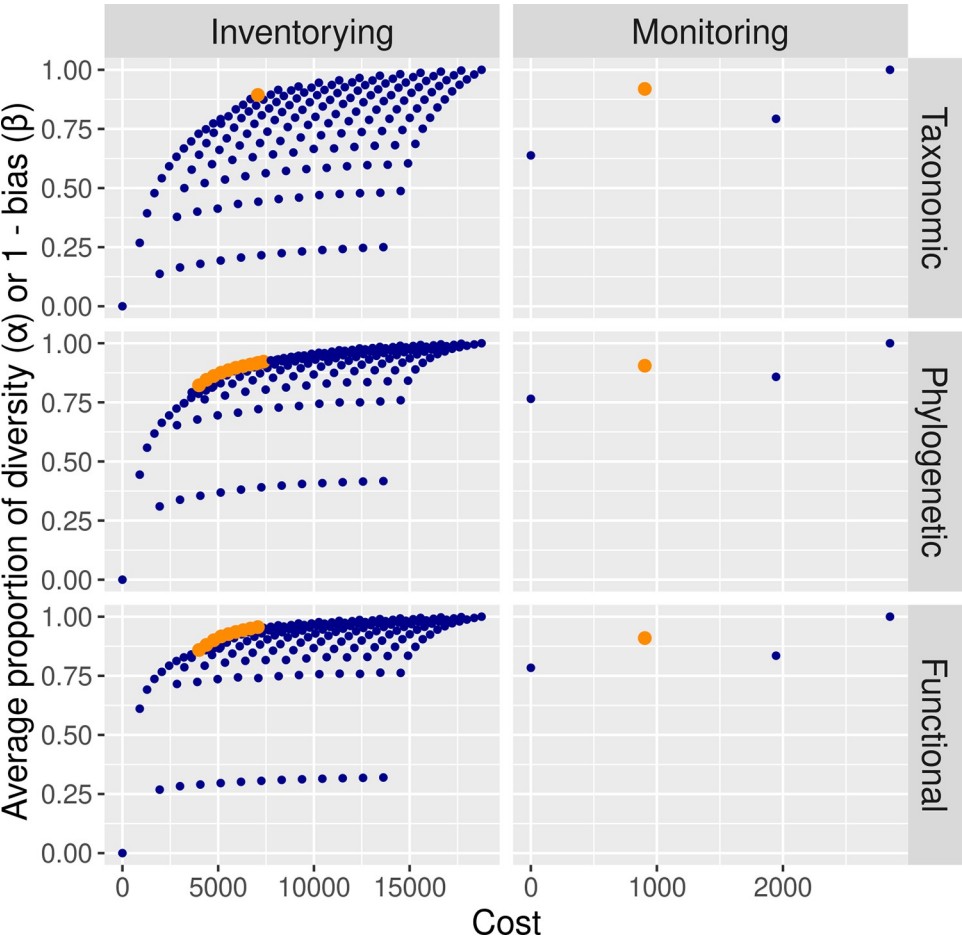

**Fig 2. Bat inventorying and monitoring optimization.** Relationship between the costs of implementing the inventorying scheme and the resulting α-diversity and bias in the assessment of the β-diversity. The results are shown for taxonomic, phylogenetic and functional diversity of Amazonian bats. For inventorying, α-diversity is scaled from 0 (no diversity) to 1 (diversity recorded in the scheme containing all the methods), and for monitoring, results are presented as 1—bias, so, the greater the value, the lower the bias. The dots represent all the possible combinations of methods used to survey bats. We used mist-nets and acoustic recorders. The most cost-effective schemes are in orange.

recorders, we achieve 94.8% of the PD with 49.0% of the cost of the full scheme (Fig 2). Therefore, the decision considering this trade-off between higher diversity and higher costs must be made taking into account the resources available for the study. Similarly, for functional α-diversity, the most cost-effective solution is subjective and may include four to 12 mist-netting sites combined with one acoustic recording site, which returned 85.9% to 95.6% of the FD with 21.3% to 37.7% of the costs of the full scheme (Fig 2).

For biodiversity monitoring, the best sampling scheme is to sample only with mist-nets for TD, PD, and FD, which results in biases <10% (Fig 2).

## Discussion

Here we propose a method to optimize and standardize the inventory and monitoring of taxonomic, phylogenetic and functional diversity, and further introduce the definitions of α-sampling and β-sampling. We demonstrate the method using Iberian spiders and Amazonian bats as case-studies. This same method is applicable to any taxon at any region, as long as sampling

can be done using a combination of methods. In fact, a first test for monitoring of TD was already proposed for Macaronesian spiders, beetles and bryophytes [64], proving the usefulness of the concept for different taxa.

In complement to the well-informed opinions of researchers, this approach allows the proposed protocols to meet six requirements that facilitate their widespread use: (1) Suitability—the methods and effort should be designed for the target organisms and the questions to be answered, and should sample a substantial and known part of the community; (2) Efficiency—the chosen methods and their proportions should gather the best information with the least effort; (3) Feasibility—the methods and effort employed must be adjusted to available resources (human, financial, time); (4) Flexibility—different sampling occasions or projects will have different resources available and the protocols should consider these limitations; (5) Transparency—if the logic behind the inventory design is clearly explained, different teams will be better able to replicate it; and (6) Accountability—the results from a chosen protocol must be adequate to support a posteriori evaluation and review [23].

The algorithm proposed here can support researchers and environmental managers to choose optimal sampling schemes for inventory and monitoring. However, the choice of the optimal sampling scheme may depend on project goals, and the availability of money, time, and human resources. Although here we only considered money as a cost parameter to support the choice of the best sampling scheme, the demand for time or human resources can also be included as cost parameters in the algorithms. The choice of the most adequate cost parameter depends on which factor is more limiting to the inventory/monitoring project. In some cases, the researchers or environmental managers may even choose sub-optimal sampling schemes if the resource availability is very limited. In this case, the proposed algorithm is useful to indicate how much of the diversity is being assessed in the inventory or the amount of bias that is being introduced in the monitoring. We chose to focus on trees as representations for both PD and FD because they allow the comparison of TD, PD and FD under a common framework [23,44,45]. This optimization method can however be adapted to other representations, such as multidimensional kernel-density hypervolumes, a common representation for FD [29,65–67].

For Iberian spiders, the monitoring protocol we propose was delineated using methods targeting different micro-habitats (Table 2). This probably explains why a combination of several different methods is optimal, as different components of the assemblages should be targeted. Different habitats could require a different combination of methods, depending on, among others, existence of an arboreal stratum. Optimizing β-sampling may therefore depend on the goals and, although ideally the same methods should be used in different settings, this is not always possible and optimization could be done for each biome, region and/or habitat. This is also supported by the fact that β-diversity is usually unitless, so it is possible to compare protocols using different methods and efforts. Contrary to inventorying, which can be, and is being done, at a global level using very similar methodology, allowing comparisons at large to global scales, monitoring is often case-specific, and results context-dependent.

For Amazonian bats, the optimal sampling protocol varied between TD, PD, and FD for α-sampling. Therefore, if a project aims to use different diversity types, it is important to assess the best options for all the diversity types to avoid choosing sub-optimal protocols.

Nevertheless, ideally, before starting any kind of monitoring of a site or region, an inventory should be conducted, so that a baseline or control can be established [10]. Inventories provide the baseline that allows perceiving if changes in community composition are due to changes in the abundances of species, existing but previously rare clades or traits at a site, or to the replacement (i.e. $\beta_{repl}$), extirpation or arrival (i.e. $\beta_{rich}$) of new components to the community [31].

Uncoupling of real vs. sampling artifact related change in monitoring protocols is a hard task and deserves careful consideration. First, the data used to define the protocol is of critical

importance. It must be extensive, from multiple sites covering the entire spectrum of taxa and environmental conditions we want to monitor, to provide as good a baseline as possible. Second, we must be careful choosing the β-diversity measures used to optimize the protocol. The ones used here are much less prone to error due to undersampling than either α-diversity or other β-diversity measures [33]. Also, careful consideration should be given to the details of implementation of the methods. For example, in the case-study of Iberian spiders we always sampled 100x100m plots and these protocols are only comparable within similar sampled areas. In the case study of Amazonian bats, the choice of the best sampling scheme should depend on the number of mist-nets used in each sampling site.

Finally, one may combine simultaneous α and β-sampling. One may α-sample several reference sites and β-sample additional sites with the sole purpose of comparing them with the reference condition. This might be useful for example to test change in community composition along a disturbance gradient, test the effect of habitat margins, or study meta-population source-sink dynamics in a fragmented landscape.

## Setting up a sampling scheme

A few considerations must be made before the replication of any sampling protocol. The spatial extent and temporal scale to be applied in the future protocol must be determined in view of specific objectives and available resources. The number and location of sampling sites could, as much as possible, be consistent with some procedure common for an entire scheme. Many projects may benefit from maximizing environmental diversity covered by sampling (including habitat types), with the reasoning that environmental and biological diversity may be correlated [68,69]. This allows maximizing the overall diversity captured and, most importantly, the range of variation in biodiversity of a region. One must remember though, that if monitoring is to be done, the sites should be fixed, and ease of accessibility may be important.

Although we optimized taxonomic, phylogenetic and functional diversity separately, researchers may want to apply a single protocol. Often the reached number of samples is highly correlated as was the case for our examples. In practice, we may find a single protocol that balances the optimal number of samples per diversity dimension (e.g., using the median number of samples per method between TD, PD and FD) and that considers the logistics that are planned for its application in the field (e.g., if it is impossible to sample for more than two hours during the night, one might modify the protocol to meet this requirement).

For monitoring, the frequency and duration of the scheme is crucial. Often it is not possible to detect changes in species composition or population abundance before a relatively large time lag [70,71]. It is also important to be able to tell apart consistent change and natural fluctuations. Thus, besides specimen sampling, it is important to collect the most relevant biotic and abiotic variables. Usually habitat type, temperature, precipitation, tree and ground cover during or immediately before the time of sampling could potentially be useful to help understanding community composition and change. Precipitation is known to strongly influence the activity of many animals and collectors should therefore make an effort to only sample in appropriate conditions [72].

Importantly, all parameters used for sampling optimization should be published. This will guarantee FAIR access to data [73] and proper replication of the protocol at large spatial and temporal scales by different teams, the protocol outliving its initial proponents. Additionally, with many species and samples the algorithms might take several hours to run. We advise to first test if all code is ok with few (e.g. 10) runs and only after a testing stage do the full analyses with 1000+ runs.

In many cases, voucher specimens, or better yet entire collections, should be preserved as only this way it is possible to guarantee future comparability of projects if taxonomy changes or if further analyses are needed relying on genetic or functional data. One may for example want to sequence new genes that allow the comparison of intra-specific genetic diversity of given species between sites or years even if these comparisons were not thought of from the beginning of the project. One may also want to detect changes in functional traits such as body size driven by environmental change. All projects involving specimen sampling should therefore plan and allocate appropriate funding to specimen deposition in public collections such as in natural history museums [74].

## Conclusions

The widespread use of similar sampling protocols at global levels and regular repetition in time can potentially have a major impact on the scope and consequently usefulness of projects such as GEO-BON, RLI and LPI. But the impact goes much beyond these. Often researchers use sampling methods and effort per method with which they are more familiar with or that are specific to a given project, necessarily limited in space and time. Besides potentially low cost-effectiveness, this creates data that are not comparable with other datasets collected by different or even the same researchers for other projects. The comparable sampling of biodiversity in space and/or time allows reusing of data collected for specific purposes, potentiating a synergistic effect among different projects. This makes data useful much beyond their initial plan. We therefore strongly advocate the optimization, standardization and widespread adoption of sampling protocols for all taxa at a global level, for both inventory and monitoring of all levels of biodiversity: taxonomic, phylogenetic and functional.

## Supporting information

**S1 Table. Eighteen sampled sites in Iberian oak forests.** At two sites (with *) 320 samples were performed and these were used only for the optimization of inventorying. At 16 sites, 24 samples were performed, and these were used only for the optimization of monitoring. (PDF)

## Acknowledgments

We thank all the collectors of past and present studies that contributed to the dataset used in this paper: Luis Carlos Crespo, Rui Carvalho, Alberto de Castro, Cristina Frías-López, Pedro Humberto Castro, Clara Gaspar, Ana Filipa Gouveia, Sérgio Henriques, Eva de Mas, Paola Mazzuca, Elisa Mora, Jordi Moya, Vera Opatova, Luis Carlos Pereira, Enric Planas, Carles Ribera, Marcos Roca-Cusachs, Dolores Ruiz, Nikolaj Scharff, Jesper Schmidt, Israel Silva, Ricardo Silva, Pedro Sousa, Tamás Szüts, Vanina Tonzo, João D. Miguel, Adrià López-Baucells, Ricardo Rocha, Jorge M. Palmeirim, Isaí Castro, Bruna S. Xavier, Karen Mustin, Daniela Rato, Cledinaldo Marques, Cremilson Marques, Gustavo Silveira, Joandro Pandilha, Jackson Souza, Luís Miguel Rosalino, Maíra S. M. Godoy, Renan M. Dias, Carlos E. L. Esbérard and Cristina H. Adania.

## Author Contributions

**Conceptualization:** Pedro Cardoso, Renato Hilário.

**Data curation:** Pedro Cardoso, Miquel A. Arnedo, Nuria Macías-Hernández, William D. Carvalho, José C. Carvalho.

**Formal analysis:** Pedro Cardoso, Nuria Macías-Hernández, Renato Hilário.

**Funding acquisition:** Pedro Cardoso, Miquel A. Arnedo, William D. Carvalho.

**Investigation:** Pedro Cardoso, Miquel A. Arnedo, Nuria Macías-Hernández, José C. Carvalho, Renato Hilário.

**Methodology:** Pedro Cardoso, Renato Hilário.

**Project administration:** Pedro Cardoso.

**Resources:** Pedro Cardoso.

**Software:** Pedro Cardoso, Renato Hilário.

**Supervision:** Pedro Cardoso.

**Validation:** Pedro Cardoso, Miquel A. Arnedo, Renato Hilário.

**Visualization:** Pedro Cardoso, Renato Hilário.

**Writing – original draft:** Pedro Cardoso, Nuria Macías-Hernández, Renato Hilário.

**Writing – review & editing:** Pedro Cardoso, Miquel A. Arnedo, Nuria Macías-Hernández, William D. Carvalho, José C. Carvalho, Renato Hilário.

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
