## [Decision Letter · Decision Letter 0]

8 Apr 2024

PONE-D-24-08190Optimal inventorying and monitoring of taxonomic, phylogenetic and functional diversityPLOS ONE

Dear Dr. Cardoso,

Thank you for submitting your manuscript to PLOS ONE. After careful consideration, we feel that it has merit but does not fully meet PLOS ONE’s publication criteria as it currently stands. Therefore, we invite you to submit a revised version of the manuscript that addresses the points raised during the review process.

We look forward to receiving your revised manuscript.

Kind regards,

Petr Heneberg

Academic Editor

PLOS ONE

Journal Requirements:

   "PC has received funding from the European Union’s Horizon 2020 and Horizon Europe research and innovation programmes under grant agreements No. 861924 (SustInAfrica) and 101081964 (BioMonitor4CAP) respectively. PC was also supported by the Centre for Ecology, Evolution and Environmental Changes (cE3c, https://doi.org/10.54499/UIDB/00329/2020) and the Global Change and Sustainability Institute (CHANGE, https://doi.org/10.54499/la/p/0121/2020). MA was supported by grant 485/2012 of the Organismo Autónomo de Parques Nacionales (Ministerio de Agricultura, Alimentación y Medio Ambiente). NMH was supported by the H2020-MSCA-IF-2015 Programme (BIODIV ISLAND-CONT project, Grant no. 706482). WDC is supported by ‘Ayudas Maria Zambrano’ (CA3/RSUE/2021-00197), funded by the Spanish Ministry of Universities. Open access funded by Helsinki University Library."

Reviewers' comments:

Reviewer's Responses to Questions

**Comments to the Author**

1. Is the manuscript technically sound, and do the data support the conclusions?

Reviewer #1: Yes

Reviewer #2: Partly

Reviewer #3: Partly

2. Has the statistical analysis been performed appropriately and rigorously? 

Reviewer #1: Yes

Reviewer #2: I Don't Know

Reviewer #3: No

3. Have the authors made all data underlying the findings in their manuscript fully available?

Reviewer #1: Yes

Reviewer #2: No

Reviewer #3: Yes

4. Is the manuscript presented in an intelligible fashion and written in standard English?

Reviewer #1: Yes

Reviewer #2: Yes

Reviewer #3: Yes

5. Review Comments to the Author

Reviewer #1: This is an interesting paper, which I generally enjoyed reading. In view of the biodiversity crisis, optimising sampling designs for cost-effective species inventories and monitoring remains a key priority and I found the authors’ approach of jointly targeting all three diversity dimensions appealing. I think the paper has great potential to make a valuable contribution to the literature on the subject. Although the clarity of the text could be improved in places, overall, I think the manuscript is quite well written and the results are adequately presented. I am not entirely convinced, however, that much is gained by the inclusion of all three case studies. I think that one or two well-explained examples would probably be sufficient to illustrate the general approach. That’s because in the end all of these examples are just that – examples – out of an enormous universe of potential case studies, each with its own sets of constraints. I detail a number of other, mostly minor, issues below for the authors’ perusal.

l. 69: “biased” instead of “concentrated”

l. 82: I’d use “occur” rather than “live”

l. 86: Why would a species inventory, if e.g. focussed on the local scale, require sampling of an entire region? Yes, if there’s a regional focus, but that’s surely not always the case, so this clearly is scale-dependent.

l. 90: Although community composition is just one (out of many) possible state variables in a monitoring program…

l. 103: hereafter

Table 1: You state that monitoring focuses on selected taxa. While this is true in many cases, it clearly depends on the state variable being monitored. If the state variable is, for instance, species richness, monitoring still relies on information on ideally most species present in a community. I would also say that in terms of geographic scope, both inventories AND monitoring programmes can target different spatial scales, from local, regional to global.

l. 130-132: It would be good to provide a reference here.

l. 266: Vespertilionidae

l. 270/271: presumably these were two consecutive nights of sampling and these were nets set at ground level?

l. 277-279: I am not sure I can follow your calculations here for fixed costs. You say mist nets cost $360 each (which seems like a lot) and poles (presumably a set of two?) cost $160. So, that’s $520 per net and thus $9360 for 18 nets! Please double check your cost calculations.

l. 281: Also, another important consideration in mist-netting surveys is whether to sample on consecutive nights (which is less ideal due to declining capture rates as a result of net shyness but often logistically easier and thus more cost-efficient) or to space repeat visits to the same site sufficiently apart in time (which adds to the cost of sampling but increases detection rates), cf. Marques et al. (2013). Optimizing Sampling Design to Deal with Mist-Net Avoidance in Amazonian Birds and Bats. PLoS ONE, 8(9), e74505)

l. 285 & 288: recorders instead of records

l. 287 & 290: recordings

l. 289/290: How exactly did you arrive at the value of $864? Analysis of echolocation calls of tropical bats is very time-consuming, requires expert knowledge and to this date cannot be reliably achieved using automatic classifiers alone – manual verification of a considerable fraction of the recordings is pivotal (cf. López-Baucells et al. (2019) Stronger together: Combining automated classifiers with manual post-validation optimizes the workload vs reliability trade-off of species identification in bat acoustic surveys. Ecological Informatics, 49, 45-53). So, these variable costs related to species IDying based on acoustic data will obviously depend heavily on the number of recordings obtained, can vary markedly, but will probably in most cases be substantially higher than fixed costs.

L. 306: as above, 40 consecutive days?

l. 312-315: for the acoustic recorders you considered the equipment cost under fixed costs, yet here you consider the camera traps under variable costs. Please clarify.

l. 374: Why were the calculations of FD for bats based on only four traits compared to 11 and 8 for spiders and mammals, respectively?

l. 437: That’s a surprisingly low number of records, especially for the acoustic recorders.

l. 438: see my earlier comment re cost calculations for mist netting (l. 277).

l. 452/453: No acoustic recorders and only mist nets for alpha-FD inventorying: This result is rather counterintuitive given that the Audiomoths recorded many unique species not captured in mist nets…This might be a reflection of the small number of traits considered for bats.

l. 502: and project goals

l. 532: I suggest rewriting to: …due to changes in the abundances of species, existing but previously rare clades or traits at a site, …

l. 537: Can you be more specific in relation to what you mean “entire spectrum”?

l. 558/559: This sentence requires some rewriting for clarity.

Reviewer #2: The ambiguity in my answers above to questions 1-4 is due to my inability to reproduce all of the analyses due to missing data tables. Specifically, data for bats and forest mammals were not available.

This paper addresses important biodiversity study design questions. It is largely very well written. Below are suggestions by line number to help increase clarity where I found it lacking. The examples were varied, potentially leading to meaningful comparisons, however, as noted above I was not able to fully explore their examples due to missing data. Results are presented succinctly. I recommend the addition of a small table (similar to Table 2), summarizing the findings for the 3 parameters and 2 mammal groups.

A misunderstanding I held upon initially reading the methods, was that the author’s optimization exercise would allow researchers to *simultaneously* optimize taxonomic, phylogenetic, and functional diversity capture and minimize bias. Instead, the methods optimize the sampling design for each parameter (TD, PD, and FD) *independently*. In every scenario, the optimal combination of methods varied by parameter. This then presents the researchers with the task of deciding which parameter to optimize. Would it be possible to extend the methods so that the simultaneous optimization is possible? I don’t imagine it is as simple as summing the maximum samples identified in each method of each parameter of an optimization (e.g., for spiders, 1 AD, 6 AN, 4 BD, 2 BN, 5 GD, 1 GN, 9 PF, 0 SD, 6 SN for a cumulative effort of 34 sample units).

Optimizing sampling so that it varies for different ecoregions, depending on habitat complexity, has the long-term negative repercussion that as habitats change, sampling is optimized only to initial conditions, and may not capture changes associated with new microhabitats or conditions. Why not optimize sampling effort for the most complex, complete situation; at sites where fewer microhabitats are present, sampling is abbreviated, but should those microhabitats appear (e.g., shrubification of grasslands), methods are a priori developed to include those new strata.

The message seems to shift a bit throughout the manuscript. Should researchers individually and somewhat subjectively optimize sampling for their specific region, ecoregion, parameter prioritization as indicated in the discussion? Or should we be aiming for a global set of generalizable protocols that can be adapted transparently in disparate regions, as the conclusions indicate? If they are advocating for the former – would more transparent publication of survey parameters allow us to bridge the gap between studies that choose different optimizations?

The supporting R code and data require some enhancements to make them functional for readers, detailed below. In brief, quite a few supporting data tables are not provided, and quite a bit of the script commenting is in Portuguese.

.R file, data files, and ability for the reader to reproduce the analyses

Given this is an English language journal, having the code comments in English is recommended. Or even better – keep the existing partly Portuguese version and create a second English language version!

It would also be helpful if the authors provided additional definitions/comments to help users follow their example. The text becomes less clear the deeper the reader goes into the script, and I find it unlikely that a reader could apply their methods easily even if all the data were provided, as the script stands.

The authors indicate that all code to replicate the analyses is deposited, but I did not find that to be true. For real reproducibility, all data tables required to rerun the code need to be provided. Right now the reader is not provided with some tables within the code, e.g,. dados_optim.csv, functional traits.csv, Morcegos_optim_func.csv for the mammal analyses.

Only the spider functional data are provided in the link https://doi.org/10.6084/m9.figshare.8320004.v3. No mammal functional data are provided. If possible, all supporting data table should be provided at the github link, with reference to their source publications as required. A large part of replicating analyses is understanding how data should be structured, so these example data tables are critically important to readers.

At present there is also reference to a personal Onedrive within the code.

I was only able to partially replicate the spider analyses, and it took my computer quite a few hours to run through the just the first part of analyses (still chugging after 5 hours). Perhaps some comments or warning on the computation requirements would be useful.

Questions by line in the manuscript

35 – abstract – unclear -what do you mean by “claims for”? perhaps “requires”?

64,65 – simplify – if this era is incommensurable, how do we then compare it to other extinction eras?

67 – provide a reference for statement

105 – table 1 – some of these requirements vary by taxon and study so this comes across as too simplistic or formulaic. For example, non-motile, perennial species don’t require considerations of seasonality. The cutoff between >50% and <50% seems arbitrary. And surely the scope of inventory is as dependent on the study or aspect of biodiversity as monitoring, not necessarily global.

111 – my preference would be to use the full terms throughout rather than the abbreviations, but I realize these abbreviations have been used by previous studies as well – so just a preference. It would also be helpful to define them briefly the first time they are introduced.

117 – what do the authors mean by heavy users? Almost every country in the world monitors some aspect of biodiversity. NatureServe compiles conservation ranks across many jurisdictions. The UN has a variety of indices and reports on biodiversity. Given the depth of policy and research in this area, I recommend more literature review or a better definition of what they considered a heavy user. Or even more simply, just treat the three main examples as just that – examples.

135-136 – our local experience suggests that those relatively large, charismatic, mobile megafauna can have some of the most variable data, making tracking trend very challenging. And we have some of the best resources in the world. I recommend you modify that statement.

146 – change “implied” to “incurred” or something similar

153 – remove “In an ideal world, sampling biodiversity would imply no costs. In the real world” and start the next sentence with “Any”

169 – change “data is” to “data are”

257-258 – given the authors know the fixed and presumably variable costs, I’d encourage them to show us their methods full potential and consider them in their full complexity rather than simplify their example. I’m unclear why they simplified the spider example given their goals. If the idea is to show an example for researchers that don’t have the full suite of valuations, then I recommend they state that.

330 – change “concatenate matrix” to “concatenated matrix”

348 – change “genetic data was” to “genetic data were”. In general check that data are treated as plural consistently throughout.

429 – I could be misunderstanding? I disagree with the authors statement that “For β-diversity, the optimal combination of six samples for all measures was similar (Table 2). TD and PD seem to require sampling methods that FD does not for optimization. Ie, TD requires 2 samples of night branch beating; PD requires 3 nights of aerial surveys. I see more agreement in methods for alpha diversity, but it is not apparent to me for beta. Put another way, optimization of TD required 2/6 methods that were not required for PD or FD.

I’m also curious – why is only TD calculated in a constrained manner? Could you also constrain PD and FD in these analyses?

With regard to the spider results, in the methods you discuss a nested optimization (line 59), but I don’t see results reported for this analysis.

478-479 – this to me was a fascinating finding, given the programs I see that are increasingly adopting cameras over tracking. In fact I haven’t heard of a program moving away from camera trapping, despite the issues you address here (high cost, low capture, subsequent high variability). I wonder if this does not merit highlighting in the abstract and a few sentences in the introduction.

498 – criterion 5 – by “process to reach them” do you mean “process to compare them”? These could be slightly revised for clarity, following the more fulsome descriptions in Cardoso 2009.

523-525 – I generally appreciate the distinction the authors draw between inventorying and monitoring, but I disagree with this statement. Should we not be arguing for the same kind of standardization and optimization for monitoring worldwide so that monitoring results can be compared in disparate regions? How else will we know which regions need further conservation effort, given the larger effort required and greater time spans between the more singular inventorying efforts? That standardization may actually be more important for monitoring, given if an inventory is done well, you reach a high level of completeness, and differences in method become more about efficiency than the actual resultant data.

Reviewer #3: The research question and topic are of great importance and relevance. All approaches that seek the interoperability of scientific studies are of great relevance at this time within the framework of the global objectives of conservation and sustainable use of biodiversity. However, the manuscript is scattered, covers too many topics superficially, and is considered to use some terms lightly. Firstly, it is considered that this study can be much more focused on the issue of optimizing techniques for sampling the number of species (alpha diversity). Regarding monitoring, it is considered that the concept is not correctly used, given that it is not clearly defined and the main characteristic of a monitoring exercise is omitted, which is to answer a specific question about the change in a variable or characteristic of biodiversity. Beta diversity, or beta sampling (a concept that is not entirely clear in the manuscript) is considered a macroecological pattern whose measurement has several debates still open. This makes the selection of a sampling protocol more complex as well as the definition of characteristics required by the optimization algorithm proposed in the manuscript. Additionally, monitoring on biodiversity elements has been reported to be a minimum of five to ten years, so the proposed examples do not seem to meet this criterion. What is observed in the manuscript are case studies of inventory exercises of specific species at specific times and places and it is considered that they should be presented as such and not as a generalized method to optimize techniques for inventories and monitoring of taxonomic phylogenetic and functional diversity.

On the other hand, it is suggested to accompany the description of the sampling protocols with geographical information on the distribution of the samples according to the protocol, and the area of interest that is sampled. This is to give more clarity to the scope of the results of the optimization algorithm and to give a better guide so that the methodology can be replicated.

Finally, the justification for the study is considered to be light: first, it is incorrect to continue using the term "developing countries" to refer to countries in the global south. Second, in the global south, where there is the highest concentration of species, there are great capacities, both human and financial, to develop inventories and biodiversity monitoring projects. Saying that conducting this study is important "considering that the majority of biodiversity occurs in developing countries, where there are few resources (both human and financial) to carry out inventories and monitoring", is considered an incorrect statement and that it can be perpetuating and increasing the gaps between the global north and south in the work for the study and conservation of biodiversity.

6. PLOS authors have the option to publish the peer review history of their article (what does this mean?). If published, this will include your full peer review and any attached files.

Reviewer #1: No

Reviewer #2: No

Reviewer #3: No

---

## [Author Response · Author response to Decision Letter 0]

28 May 2024

Reviewer #1: This is an interesting paper, which I generally enjoyed reading. In view of the biodiversity crisis, optimising sampling designs for cost-effective species inventories and monitoring remains a key priority and I found the authors’ approach of jointly targeting all three diversity dimensions appealing. I think the paper has great potential to make a valuable contribution to the literature on the subject. Although the clarity of the text could be improved in places, overall, I think the manuscript is quite well written and the results are adequately presented. I am not entirely convinced, however, that much is gained by the inclusion of all three case studies. I think that one or two well-explained examples would probably be sufficient to illustrate the general approach. That’s because in the end all of these examples are just that – examples – out of an enormous universe of potential case studies, each with its own sets of constraints. I detail a number of other, mostly minor, issues below for the authors’ perusal.

Authors: We thank the reviewer for the comments and decided to exclude one of the case studies on the small mammals.

R1: l. 69: “biased” instead of “concentrated”

A: Done.

R1: l. 82: I’d use “occur” rather than “live”

A: Agreed, done.

R1: l. 86: Why would a species inventory, if e.g. focussed on the local scale, require sampling of an entire region? Yes, if there’s a regional focus, but that’s surely not always the case, so this clearly is scale-dependent.

A: We meant the region of interest, no scale intended. This was now clarified.

R1: l. 90: Although community composition is just one (out of many) possible state variables in a monitoring program…

A: That is correct, we are limiting our analyses to community composition hence the text. This is now clarified.

R1: l. 103: hereafter

A: Corrected.

R1: Table 1: You state that monitoring focuses on selected taxa. While this is true in many cases, it clearly depends on the state variable being monitored. If the state variable is, for instance, species richness, monitoring still relies on information on ideally most species present in a community. I would also say that in terms of geographic scope, both inventories AND monitoring programmes can target different spatial scales, from local, regional to global.

A: That is correct, we meant to say that monitoring is often focused on selected taxa. To avoid confusion, and as both might encompass multiple scales, we decided to delete this line.

R1: l. 130-132: It would be good to provide a reference here.

A: We added a general reference as it is hard to prove the negative, only a number of studies that did not find correlations between limited numbers of taxa.

R1: l. 266: Vespertilionidae

A: Corrected

R1: l. 270/271: presumably these were two consecutive nights of sampling and these were nets set at ground level?

A: We clarified that the surveys in the same site were non-consecutive and were carried out at least 30 days apart.

R1: l. 277-279: I am not sure I can follow your calculations here for fixed costs. You say mist nets cost $360 each (which seems like a lot) and poles (presumably a set of two?) cost $160. So, that’s $520 per net and thus $9360 for 18 nets! Please double check your cost calculations.

A: We used the word ‘each’ erroneously in this sentence. Indeed, these were the costs of 18 mist-nets and 20 poles. Each mist-net costs about $20 and each pole costs about $8. We have corrected the text now. However, note that the total fixed cost that we used in the analysis is correct.

R1: l. 281: Also, another important consideration in mist-netting surveys is whether to sample on consecutive nights (which is less ideal due to declining capture rates as a result of net shyness but often logistically easier and thus more cost-efficient) or to space repeat visits to the same site sufficiently apart in time (which adds to the cost of sampling but increases detection rates), cf. Marques et al. (2013). Optimizing Sampling Design to Deal with Mist-Net Avoidance in Amazonian Birds and Bats. PLoS ONE, 8(9), e74505)

A: We clarified that the surveys in the same site were non-consecutive and were carried out at least 30 days apart.

R1: l. 285 & 288: recorders instead of records

A: Corrected.

R1: l. 287 & 290: recordings

A: Corrected

R1: l. 289/290: How exactly did you arrive at the value of $864? Analysis of echolocation calls of tropical bats is very time-consuming, requires expert knowledge and to this date cannot be reliably achieved using automatic classifiers alone – manual verification of a considerable fraction of the recordings is pivotal (cf. López-Baucells et al. (2019) Stronger together: Combining automated classifiers with manual post-validation optimizes the workload vs reliability trade-off of species identification in bat acoustic surveys. Ecological Informatics, 49, 45-53). So, these variable costs related to species IDying based on acoustic data will obviously depend heavily on the number of recordings obtained, can vary markedly, but will probably in most cases be substantially higher than fixed costs.

A: The reviewer is right and we considered that we need 18 days of work of an expert to identify the vocalizations of each site (6 recorders), paying $48 per work day. We clarified this in the text now.

R1: L. 306: as above, 40 consecutive days?

A: This case study was now deleted.

R1: l. 312-315: for the acoustic recorders you considered the equipment cost under fixed costs, yet here you consider the camera traps under variable costs. Please clarify.

A: This case study was now deleted.

R1: l. 374: Why were the calculations of FD for bats based on only four traits compared to 11 and 8 for spiders and mammals, respectively?

A: Other bat studies also include wing morphology (wing load and wing aspect) in the calculation of functional diversity (e.g. Farneda et al. 2018; Carvalho et al. 2021). However, we could not use these traits because they are not available for several insectivorous species. Other functional traits such as physiological traits and home range, are also unavailable for most species and therefore are not commonly used in studies. Nevertheless, we have now included an additional trait: vertical stratification. So, now we have 5 functional traits for bats, which are commonly used in studies with Phyllostomidae, Verspertilionidae, Molossidae, and Emballonuridae. 

References:

Carvalho WD, Mustin K, Farneda FZ, de Castro IJ, Hilário RR, Martins AC, Miguel JD, da Silva Xavier B, de Toledo JJ. Taxonomic, functional and phylogenetic bat diversity decrease from more to less complex natural habitats in the Amazon. Oecologia. 2021;197:223-239.

Farneda FZ, Rocha R, López-Baucells A, Sampaio EM, Palmeirim JM, Bobrowiec PE, Grelle CE, Meyer CF. Functional recovery of Amazonian bat assemblages following secondary forest succession. Biological Conservation. 2018;218:192-199.

R1: l. 437: That’s a surprisingly low number of records, especially for the acoustic recorders.

A: Indeed, we mistakenly included lower record numbers here, especially for bat passes. We have corrected these numbers now. The capture rates in this study case are similar to other studies in the same region (e.g. Carvalho et al. 2021; Carvalho et al. 2022)

References:

Carvalho WD, Mustin K, Farneda FZ, de Castro IJ, Hilário RR, Martins AC, Miguel JD, da Silva Xavier B, de Toledo JJ. Taxonomic, functional and phylogenetic bat diversity decrease from more to less complex natural habitats in the Amazon. Oecologia. 2021;197:223-39.

Carvalho WD, Rosalino LM, da Silva Xavier B, De Castro IJ, Hilário R, Marques TM, De Toledo JJ, Vieira MV, Palmeirim JM, Mustin K. The relative importance of forest cover and patch-level drivers for phyllostomid bat communities in the Amazonian Savannas. Landscape Ecology. 2023;38(1):117-30.

R1: l. 438: see my earlier comment re cost calculations for mist netting (l. 277).

A: We have corrected the text, clarifying the costs of mist-nets and poles

R1: l. 452/453: No acoustic recorders and only mist nets for alpha-FD inventorying: This result is rather counterintuitive given that the Audiomoths recorded many unique species not captured in mist nets…This might be a reflection of the small number of traits considered for bats.

A: The reviewer is right. Given that we included an additional functional trait (vertical stratification), the result has changed and the optimal schemes now include one site with acoustic recorders.

R1: l. 502: and project goals

A: Agreed, added.

R1: l. 532: I suggest rewriting to: …due to changes in the abundances of species, existing but previously rare clades or traits at a site, …

A: Done.

R1: l. 537: Can you be more specific in relation to what you mean “entire spectrum”?

A: Added.

R1: l. 558/559: This sentence requires some rewriting for clarity.

A: We hope it is clear now.

Reviewer #2: The ambiguity in my answers above to questions 1-4 is due to my inability to reproduce all of the analyses due to missing data tables. Specifically, data for bats and forest mammals were not available.

A: We made all the data available now.

R2: This paper addresses important biodiversity study design questions. It is largely very well written. Below are suggestions by line number to help increase clarity where I found it lacking. The examples were varied, potentially leading to meaningful comparisons, however, as noted above I was not able to fully explore their examples due to missing data. Results are presented succinctly. I recommend the addition of a small table (similar to Table 2), summarizing the findings for the 3 parameters and 2 mammal groups.

A: We think such a table would be redundant with the text and the plots, which already describe the optimal choices and the resulting diversity. Also, the bat sampling scheme is simpler than the spider sampling scheme, with only two methods, compared with 5 methods and diurnal and nocturnal sampling. Therefore, it is much easier to describe the bat results in the text, as we have done.

R2: A misunderstanding I held upon initially reading the methods, was that the author’s optimization exercise would allow researchers to *simultaneously* optimize taxonomic, phylogenetic, and functional diversity capture and minimize bias. Instead, the methods optimize the sampling design for each parameter (TD, PD, and FD) *independently*. In every scenario, the optimal combination of methods varied by parameter. This then presents the researchers with the task of deciding which parameter to optimize. Would it be possible to extend the methods so that the simultaneous optimization is possible? I don’t imagine it is as simple as summing the maximum samples identified in each method of each parameter of an optimization (e.g., for spiders, 1 AD, 6 AN, 4 BD, 2 BN, 5 GD, 1 GN, 9 PF, 0 SD, 6 SN for a cumulative effort of 34 sample units).

A: Although possible, we believe it would create some confusion. This way it is clear if there are discrepancies between the different dimensions of diversity. Moreover, there are two points to be made. First, usually different dimensions are highly correlated (even if not always and never perfectly). Second, once the optimization is completed one should carefully consider slight modifications that will help the logistics. For example, the spider protocols we are now applying throughout the world are close but not exactly any of the optimization. This helped in optimizing the logistics. Also, we did not sum the methods but did a median. This process is now explained in the discussion in a new paragraph.

R2: Optimizing sampling so that it varies for different ecoregions, depending on habitat complexity, has the long-term negative repercussion that as habitats change, sampling is optimized only to initial conditions, and may not capture changes associated with new microhabitats or conditions. Why not optimize sampling effort for the most complex, complete situation; at sites where fewer microhabitats are present, sampling is abbreviated, but should those microhabitats appear (e.g., shrubification of grasslands), methods are a priori developed to include those new strata.

A: That is a very good point. This was added to the text on the constrained optimization (start of material and methods section).

R2: The message seems to shift a bit throughout the manuscript. Should researchers individually and somewhat subjectively optimize sampling for their specific region, ecoregion, parameter prioritization as indicated in the discussion? Or should we be aiming for a global set of generalizable protocols that can be adapted transparently in disparate regions, as the conclusions indicate? If they are advocating for the former – would more transparent publication of survey parameters allow us to bridge the gap between studies that choose different optimizations?

A: Good point, we added a paragraph about replication at the end of the discussion.

R2: The supporting R code and data require some enhancements to make them functional for readers, detailed below. In brief, quite a few supporting data tables are not provided, and quite a bit of the script commenting is in Portuguese. .R file, data files, and ability for the reader to reproduce the analyses. Given this is an English language journal, having the code comments in English is recommended. Or even better – keep the existing partly Portuguese version and create a second English language version!

A: We have revised the R scripts and included all the datasets in the Github

R2: It would also be helpful if the authors provided additional definitions/comments to help users follow their example. The text becomes less clear the deeper the reader goes into the script, and I find it unlikely that a reader could apply their methods easily even if all the data were provided, as the script stands.

A: We tried our best to add comments as needed.

R2: The authors indicate that all code to replicate the analyses is deposited, but I did not find that to be true. For real reproducibility, all data tables required to rerun the code need to be provided. Right now the reader is not provided with some tables within the code, e.g,. dados_optim.csv, functional traits.csv, Morcegos_optim_func.csv for the mammal analyses.

A: These tables are now added to github.

R2: Only the spider functional data are provided in the link https://doi.org/10.6084/m9.figshare.8320004.v3. No mammal functional data are provided. If possible, all supporting data table should be provided at the github link, with reference to their source publications as required. A large part of replicating analyses is understanding how data should be structured, so these example data tables are critically important to readers.

A: These tables are now added to github.

R2: At present there is also reference to a personal Onedrive within the code.

A: We have corrected this problem now.

R2: I was only able to partially replicate the spider analyses, and it took my computer quite a few hours to run through the just the first part of analyses (still chugging after 5 hours). Perhaps some comments or warning on the computation requirements would be useful.

A: Added to the discussion, third paragraph.

R2: Questions by line in the manuscript

35 – abstract – unclear -what do you mean by “claims for”? perhaps “requires”?

A: Changed.

R2: 64,65 – simplify – if this era is incommensurable, how do we then compare it to other extinction eras?

A: Modified.

R2: 67 – provide a reference for statement

A: This sentence was deleted for simplification.

R2: 105 – table 1 – some of these requirements vary by taxon and study so this comes across as too simplistic or formulaic. For example, non-motile, perennial species don’t require considerations of seasonality. The cutoff between >50% and <50% seems arbitrary. And surely the scope of inventory is as dependent on the study or aspect of biodiversity as monitoring, not necessarily global.

A: We agree, modified the table accordingly.

R2: 111 – my preference would be to use the full terms throughout rather than the abbreviations, but I realize these abbreviations have been used by previous studies as well – so just a preference. It would also be helpful to define them briefly the first time they are intro

---

## [Decision Letter · Decision Letter 1]

24 Jun 2024

PONE-D-24-08190R1Optimal inventorying and monitoring of taxonomic, phylogenetic and functional diversityPLOS ONE

Dear Dr. Cardoso,

Thank you for submitting your manuscript to PLOS ONE. After careful consideration, we feel that it has merit but does not fully meet PLOS ONE’s publication criteria as it currently stands. Therefore, we invite you to submit a revised version of the manuscript that addresses the points raised during the review process.

We look forward to receiving your revised manuscript.

Kind regards,

Petr Heneberg

Academic Editor

PLOS ONE

Journal Requirements:

Reviewers' comments:

Reviewer's Responses to Questions

**Comments to the Author**

1. If the authors have adequately addressed your comments raised in a previous round of review and you feel that this manuscript is now acceptable for publication, you may indicate that here to bypass the “Comments to the Author” section, enter your conflict of interest statement in the “Confidential to Editor” section, and submit your "Accept" recommendation.

Reviewer #1: All comments have been addressed

Reviewer #2: All comments have been addressed

Reviewer #3: All comments have been addressed

2. Is the manuscript technically sound, and do the data support the conclusions?

Reviewer #1: Yes

Reviewer #2: (No Response)

Reviewer #3: Yes

3. Has the statistical analysis been performed appropriately and rigorously? 

Reviewer #1: Yes

Reviewer #2: (No Response)

Reviewer #3: Yes

4. Have the authors made all data underlying the findings in their manuscript fully available?

Reviewer #1: Yes

Reviewer #2: (No Response)

Reviewer #3: Yes

5. Is the manuscript presented in an intelligible fashion and written in standard English?

Reviewer #1: Yes

Reviewer #2: (No Response)

Reviewer #3: Yes

6. Review Comments to the Author

Reviewer #1: The authors have satisfactorily addressed all of my comments and concerns, and I found the manuscript to be considerably improved. I congratulate the authors on this nice contribution.

Reviewer #2: (No Response)

Reviewer #3: I believe that it is important to define from the beginning of the methodology what bias is.

On line 228, the bibliographic reference must be placed again for the work where the location and date data of the studies are found.

In the discussion, line 430, I suggest reconsidering the statement about the subjective opinion of the researchers, given that I consider that this is a factor that can certainly always have a very positive impact on the development of sampling and studies in general on biodiversity. I do not think it should be rejected and I think it can complement the proposal of the manuscript.

7. PLOS authors have the option to publish the peer review history of their article (what does this mean?). If published, this will include your full peer review and any attached files.

Reviewer #1: No

Reviewer #2: No

Reviewer #3: No

---

## [Author Response · Author response to Decision Letter 1]

28 Jun 2024

Reviewer #3: I believe that it is important to define from the beginning of the methodology what bias is.

Authors: This is now defined in lines 153-154.

R3: On line 228, the bibliographic reference must be placed again for the work where the location and date data of the studies are found.

A: Re-reading this we think this sentence is not needed here as the studies were not presented yet, decided to remove it.

R3: In the discussion, line 430, I suggest reconsidering the statement about the subjective opinion of the researchers, given that I consider that this is a factor that can certainly always have a very positive impact on the development of sampling and studies in general on biodiversity. I do not think it should be rejected and I think it can complement the proposal of the manuscript.

A: Agreed and sentence was modified accordingly.

---

## [Editor Report · Decision Letter 2]

2 Jul 2024

Optimal inventorying and monitoring of taxonomic, phylogenetic and functional diversity

PONE-D-24-08190R2

Dear Dr. Cardoso,

We’re pleased to inform you that your manuscript has been judged scientifically suitable for publication and will be formally accepted for publication once it meets all outstanding technical requirements.

Kind regards,

Petr Heneberg

Academic Editor

PLOS ONE
---

## [Editor Report · Acceptance letter]

23 Jul 2024

PONE-D-24-08190R2 

PLOS ONE

Dear Dr. Cardoso, 

I'm pleased to inform you that your manuscript has been deemed suitable for publication in PLOS ONE. Congratulations! Your manuscript is now being handed over to our production team.

Kind regards, 

on behalf of

Dr. Petr Heneberg 

Academic Editor

PLOS ONE